# Resveratrol and beyond: The Effect of Natural Polyphenols on the Cardiovascular System: A Narrative Review

**DOI:** 10.3390/biomedicines11112888

**Published:** 2023-10-25

**Authors:** Roland Gál, Róbert Halmosi, Ferenc Gallyas, Michael Tschida, Pornthira Mutirangura, Kálmán Tóth, Tamás Alexy, László Czopf

**Affiliations:** 1Division of Cardiology, 1st Department of Medicine, Medical School, University of Pecs, 7624 Pecs, Hungary; gal.roland@pte.hu (R.G.); halmosi.robert@pte.hu (R.H.); toth.kalman@pte.hu (K.T.); 2Szentágothai Research Centre, University of Pecs, 7624 Pecs, Hungary; 3Department of Biochemistry and Medical Chemistry, University of Pecs, 7624 Pecs, Hungary; ferenc.gallyas@aok.pte.hu; 4Medical School, University of Minnesota, Minneapolis, MN 55455, USA; tschi226@umn.edu; 5Department of Medicine, University of Minnesota Medical School, Minneapolis, MN 55455, USA; mutir001@umn.edu; 6Department of Medicine, Division of Cardiology, University of Minnesota, Minneapolis, MN 55455, USA; alexy001@umn.edu

**Keywords:** polyphenols, resveratrol, oxidative stress, inflammation, cardiovascular diseases

## Abstract

Cardiovascular diseases (CVDs) are among the leading causes of morbidity and mortality worldwide. Unhealthy dietary habits have clearly been shown to contribute to the development of CVDs. Beyond the primary nutrients, a healthy diet is also rich in plant-derived compounds. Natural polyphenols, found in fruits, vegetables, and red wine, have a clear role in improving cardiovascular health. In this review, we strive to summarize the results of the relevant pre-clinical and clinical trials that focused on some of the most important natural polyphenols, such as resveratrol and relevant flavonoids. In addition, we aim to identify their common sources, biosynthesis, and describe their mechanism of action including their regulatory effect on signal transduction pathways. Finally, we provide scientific evidence regarding the cardiovascular benefits of moderate, long-term red wine consumption.

## 1. Introduction

According to the World Health Organization (WHO) definition, the term “cardiovascular diseases” (CVDs) encompasses a variety of disorders including coronary artery disease (CAD), stroke, peripheral artery disease (PAD), hypertension, cerebrovascular disease, and heart failure (HF) [1,2,3]. Not only do they represent one of the leading causes of morbidity in countries with western lifestyles, but also contributed to 17.9 million deaths in 2019 with this number expected to exceed 23.6 million annually by 2030.

A variety of risk factors, including hypertension, dyslipidemia, diabetes mellitus, obesity, smoking, alcohol consumption, sedentary lifestyle, and unhealthy diet have been shown to contribute to the development and progression of CVDs [3]. These promote endothelial cell dysfunction, oxidative stress, smooth muscle cell, and fibroblast proliferation, and the transition of macrophages to foam cells. The resultant chronic inflammation ultimately leads to atherosclerotic plaque development across the vascular bed [4]. In addition, the complex interaction between chronic inflammation, oxidative stress, lipid metabolism, and endothelial dysfunction plays a crucial role in the development of HF [4,5,6].

Lifestyle modifications, such as improving dietary habits, are traditionally considered primary determinants of cardiovascular (CV) health. The potential benefits of dietary interventions and nutrient supplementation in improving CV outcomes have been investigated extensively over the past decades. The Mediterranean diet represents a special historical nutrition pattern consumed by populations living in the Mediterranean Basin. As demonstrated by several clinical trials in recent years, prolonged and strict adherence to this diet may reduce the incidence of CVDs by improving the CV risk profile, primarily due to its anti-inflammatory and antioxidant effects [7,8]. There is evidence that modern healthy dietary habits share many similarities with the Mediterranean diet, including the high intake of vegetables, fruits, legumes, seeds, nuts, whole grains, dairy foods, fish, seafood, and vegetable oils (e.g., olive oil) combined with the limited consumption of sweets, soft drinks, and red meat [9,10,11]. While heavy alcohol use can be detrimental, the regular but modest consumption of red wine, which is part of the Mediterranean diet, has well-documented health benefits [12]. As published by Renaud et al. in the early 1990s, the French population has a lower overall incidence and mortality from CVDs, despite a diet relatively rich in saturated fat compared to other developed countries. This observation, also known as the “French paradox”, may be attributable to the higher red wine consumption per capita in France [13].

In addition to the minerals and vitamins found in healthy diets, the role of plant-derived ingredients is also important to mention, such as natural polyphenols. These bioactive phytochemicals can be found in a wide variety of plants (e.g., fruits, vegetables, seeds, nuts), and are also present in wine, tea, spices, oils, and chocolate. They act as natural UV filters and protect from a variety of biotic (e.g., infection) and abiotic stresses (e.g., oxidative, or toxic damages, injury) owing to their antioxidant/anti-inflammatory activities [14,15,16,17,18,19]. Polyphenols also contribute to the flavor, taste, and color of plants. Their dietary intake has seen an increase in recent years with the concomitant decrease in the risk of chronic and age-related disorders, such as cancer, neurodegenerative conditions, inflammation, and CV diseases. Especially in recent years, most consumers prefer using natural ingredients as additives because of their broad availability and safety profile. Grape seed and olive oil extracts rich in polyphenols may be used as food additives [20,21].

A large number of pre-clinical and clinical trials performed over the past 25 years have provided compelling evidence of the beneficial CV effects of polyphenols [22,23,24,25,26,27,28]. However, their exact mechanism of action is yet to be fully elucidated. The most unique feature of this article is that it provides an extensive review of the favorable CV effects of polyphenols from the molecular mechanisms to the clinical data. It focuses primarily on the most relevant natural polyphenols found in red wine, including stilbenes and flavonoids. We summarize their pharmacological properties, and biopharmaceutical classification, describe their main mechanisms of action, and how they may affect cellular signal transduction pathways. We briefly discuss findings from the most relevant pre-clinical and clinical trials with a focus on CV prevention. In addition, this manuscript covers topics and areas that have not been reviewed prior, such as CV outcome trials and the effect of polyphenols on CV mortality. We trust that our review will be beneficial for clinicians and scientists planning future research in the area of polyphenols.

The well-known search engine, PubMed was used to identify literature relevant to this review. The key words included the names of the included compounds, in addition to the terms “cardiovascular”, “cardiac” or “heart”. As the relevant full-text articles were reviewed, the reference section of each manuscript was further reviewed to identify potentially relevant papers. Duplicate articles, as well as small studies with results not relevant to the current paper, were excluded. Not aiming to provide a systematic meta-analysis, the format of narrative review was selected to summarize findings from relevant pre-clinical studies and human clinical trials.

## 2. Classification, Biosynthesis, and Metabolism of Polyphenols

### 2.1. Classification and Biosynthesis of Polyphenols

The classification of polyphenols is based on the number of phenolic rings and the structural elements attached to the rings. The majority are derived through the secondary metabolism of L-phenylalanine via the phenylpropanoid pathway. Alternatively, they can be synthesized through the shikimate and/or polyketide pathways, therefore approximately 10,000 structural variants may exist [14,29,30]. The bioactive polyphenols are divided into four major subclasses: (1) flavonoids; (2) non-flavonoid stilbenes; (3) phenolic acids; and (4) lignans. The main sources of relevant polyphenols are depicted in Figure 1 [31,32].

Stilbenes, including the most widely investigated resveratrol (RES), are produced via the general phenylpropanoid pathway. They contain two phenyl residues linked by a two-carbon methylene bridge (1,2-diphenylethylene nucleus). This basic structure can subsequently be glycosylated, methylated, or prenylated by specific enzymes. In addition to RES, other stilbenes include piceatannol, pinosylvin, pinostilbene, combretastatin, polydatin, mulberroside, and various oligostilbenes [33,34]. Overall, a relatively small fraction of habitual diets contain these compounds when compared to flavonoids and phenolic acids (peanuts, berries, grapes, and red wine) [35].

Flavonoids are composed of two aromatic rings and a heterocyclic ring with a typical C6-C3-C6 skeleton structure. They are generally classified into 7 subgroups based on the degree of oxidation, hydroxylation, glycosylation, and substitutions of the central heterocyclic ring: (1) flavonols; (2) flavones; (3) isoflavones; (4) anthocyanidins; (5) flavanones; (6) flavanols; (7) chalcones. Approximately two-third of the polyphenolic compounds are flavonoids and are ubiquitous to most plants [36,37,38].

Phenolic or phenolcarboxylic acids produced by the phenylpropanoid pathway contain aromatic phenolic rings and one carboxylic acid group in their structure. They are among the main groups of polyphenols and are present in one-third of plants. They are divided into two subclasses: (1) hydroxybenzoic acid (e.g., gallic acid); and (2) hydroxycinnamic acid (e.g., caffeic acid, ferulic acid). Their aromatic phenolic rings have the ability to lose an electron, which explains their high antioxidant activity [39,40,41].

Lignans are biphenolic compounds derived from the combination of two phenylpropanoid C6–C3 units synthetized through the shikimate pathway. The ether, lactone, or carbon bonds linked to the biphenolic ring are responsible for the biological activity of these moieties. The primary compounds that are also the most studied include secoisolariciresinol, matairesinol, and pinoresinol [42,43].

### 2.2. Metabolism of Polyphenols

The bioavailability of polyphenols is poor in general. They undergo rapid and extensive metabolism in the gastrointestinal tract and liver that alters their molecular structure. These changes are naturally responsible for the low plasma levels of intact (unmetabolized) forms [44,45] that, according to the literature, rarely exceed 1 µM after consuming 10 to 100 mg of a polyphenolic compound [46]. The main mechanisms of action of stilbenes are shown in detail in Figure 2.

Polyphenols are ingested primarily in a conjugated form and are absorbed from the stomach, intestine, and colon with the rest eliminated through feces. The metabolism of glycosylated polyphenols (O-glycosides) is initiated by oral glycosidase enzymes, which is followed by conversion into aglycone form (free form) by hydroxylation in the stomach and small intestine. This process is facilitated by intestinal microbiota (phase I metabolism) [47,48]. Aglycone polyphenols are absorbed from the lumen of the small intestine into the cytosol of enterocytes by passive diffusion or by utilizing protein carriers. However, selected polyphenols (such as hydroxycinnamic acids) are resistant to enzymatic digestion and transition unchanged into the colon, where microbiota metabolize them into aglycone form. Following absorption, aglycone polyphenols are transported to the liver via the portal vein, where they are conjugated to O-sulphate or O-glucuronide forms (phase II metabolism). Consequently, polyphenols are typically found in the circulation in conjugated forms. These are filtered by the kidneys and excreted in the urine [49,50,51]. The metabolized forms may also be biologically active, and in fact may be more active than their precursors, as is the case with RES, quercetin (Qct), catechin, and isoflavons [52,53,54,55,56,57]. Furthermore, several polyphenols can penetrate and accumulate in a variety of tissues (e.g., heart, bone, adipose tissue) both in their native and metabolized forms [58,59,60,61,62].

## 3. Stilbenes and CVDs

Although stilbenes represent one of the main classes of polyphenols, they are only present in a limited number of plants, including grapes, berries, and peanuts. Consequently, their intake through regular diet is limited with red wine representing the primary source. However, their potential benefits for human health, including CVD prevention, are significant and have been the subject of numerous pre-clinical and clinical trials. To date, over 500 molecular variants of stilbenes have been described, such as piceatannol, pinosylvin, and pinostilbene, yet 3,5,4-trihydroxystilbene (RES) is the most prominent and widely studied form [63,64]. RES exists as two geometric isomers: cis- (Z) and trans- (E) with the latter representing the biologically active form [9].

### 3.1. Mechanism of Action

Stilbenes, and particularly RES, have been extensively investigated over the past decades owing to their potent antioxidant, anti-inflammatory, anti-proliferative, anti-apoptotic and mitochondrial protective effects. Collectively, these are thought to be responsible for their derived CV benefits.

Reactive oxygen (ROS) and nitrogen species (RNS) play an important role in physiological processes (cell homeostasis, signaling pathways) when present at low concentrations. However, high ROS levels promote oxidative stress within the cells and may lead to permanent damage of critical biomolecules (lipids, proteins, DNA) [65]. Although the bioavailability and plasma concentration of stilbenes are low, several in vitro studies have demonstrated that they are important antioxidant agents. Their efficacy is explained by two separate mechanisms: (1) direct accumulation in cardiac and vascular tissues; and (2) active metabolite formation [66,67]. RES has been shown to directly scavenge excessive ROS and RNS, including superoxide (O_2_^·−^), hydroxyl radical (·OH), hydrogen-peroxide (H_2_O_2_), and peroxynitrite [68,69,70]. As reported by Leonard et al. using murine macrophage cell culture, RES effectively eliminates lipid peroxidation and DNA damage by scavenging hydroxyl, superoxide, and metal-induced radicals [63]. In subsequent experiments, it protected cardiomyocytes [71] and endothelial cells [72] against oxidative toxicity and increased the level of antioxidants. Similar results were obtained using another stilbene and RES analog, piceatannol (3,4,3′,5′-tetrahydroxy-trans-stilbene) [73,74,75,76].

In further experiments, RES was proven to attenuate ROS formation by inhibiting a range of NADPH oxidases (NOX isoforms) and by modulating the activity of the mitochondrial respiratory chain enzymes [77,78,79]. It reduces NOX isoform activity through the upregulation of SIRT1 (NAD^+^-dependent histone/protein deacetylase sirtuin 1) that, in turn: (1) prompts deacetylation of transcription factor NF (nuclear factor)-κB [80]; (2) upregulates the Forkhead box O1 (FOXO1) transcription factor-mediated synthesis of superoxide dismutase (SOD), as well as the expression of catalase (CAT) and glutathione peroxidase 1 (GPx1) [81,82,83]. Dysfunction of the endothelial nitric oxide synthase (eNOS) due to tetrahydrobiopterin (BH4) deactivation (uncoupling of eNOS) can also contribute to increased ROS production and consequent oxidative damage. RES can reverse eNOS uncoupling by upregulating GTP cyclohydrolase 1 (GCH1) expression, therefore increasing BH4 biosynthesis [84,85]. Furthermore, RES can effectively eliminate free radicals by enhancing the activity of the antioxidant defense system. According to in vitro experiments and in vivo trials, RES prompts nuclear respiratory factor (Nrf-2) activation. Nrf-2, in turn, increases cellular antioxidant content (e.g., GSH) through the upregulation of genes encoding various antioxidant enzymes: uridine-diphosphate-glucuronosyltransferases (UGTs), glutathione S-transferases, γ-glutamylcysteine synthase, and sulfotransferases [86,87].

The effective inhibition of inflammatory processes is a key function of stilbenes [88,89]. NF-κB and STAT (signal transducers and activators of transcription) transcription factors play an important role in the regulation of physiological and inflammatory processes as well as apoptosis in macrophages. The activation of these factors is mediated by TLRs (toll-like receptors—transmembrane proteins of macrophages) and by their downstream signaling pathways, which lead to increased expression of pro-inflammatory cytokines, including IL-1β, IL-6, TNF-α [90,91]. As a potent anti-inflammatory agent, RES can inhibit the expression of pro-inflammatory cytokines [92,93] by suppressing NF-кB and JAK/STAT through the activation of SIRT1 [94,95]. Yeung et al. demonstrated that SIRT1 can inhibit RelA acetylation prompting the silencing of NF-κB activity [96]. Similar results were reported with piceatannol [97,98]. Yang et al. found that RES can alleviate TLR-mediated chronic inflammation by direct downregulation of TLR4 expression [99,100].

The expression of adhesion molecules, such as VCAM-1, ICAM-1, and E-selectin, are important steps in leukocyte activation. According to Zhang et al., RES can successfully inhibit adhesion molecules by suppressing the TNF-α-induced NF-κB activation [101]. Moreover, it can decrease the expression of ICAM-1 and thus monocyte adhesion by Nrf-2 pathway stimulation [102]. In addition to inhibiting pro-inflammatory pathways, stilbenes can also increase the secretion and activation of anti-inflammatory cytokines (e.g., IL-10) and enzymes, such as heme oxygenase-1 (HO-1) [103,104,105].

COX-1 and COX-2 isoforms of the cyclooxygenase enzyme are key catalysts of prostaglandin (PG) biosynthesis and they play an important role in inflammatory processes. Vascular endothelial and smooth muscle cells serve as the major source of PG production. Arachidonic acid is released by endothelial cell membrane phospholipids with the cleavage of phospholipase A2 (PLA_2_) and then metabolized by COX enzymes to generate PGs (such as PGD_2_, PGE_2_, PGI_2_) and thromboxane (TX) A_2_. Proinflammatory cytokines (IL-1, TNF-α) play a crucial role in the activation of PG biosynthesis in states of chronic inflammation by upregulation of the NF-κB and MAPK signaling pathways. The ability of RES to inhibit the COX-1 and COX-2 enzymes can contribute to its marked anti-inflammatory (COX-2) and antiplatelet effects (COX-1) [106,107]. RES can reduce COX-2 expression by the downregulation of p65, c-Jun, Fos, and NF-κB transcription factors via SIRT1 activation, thus decreasing eicosanoid production (PGE_2_, TXA_2_) and consequently reducing inflammation. Moreover, SIRT1 can also inhibit (acetylation) the AP-1 (activated by MAPKs) transcription factors that are essential for COX-2 gene expression and for PGE2 release in inflammation as well as for endothelial proliferation [108]. In a recent human trial, RES reduced the level of pro-inflammatory cytokines (IL-1, IL-6) by inhibiting oxidative phosphorylation in leukocytes. In addition, it significantly reduced gene expression encoding B cell receptors and leukocyte extravasation signaling [66].

Stilbenes, especially RES, affect different signal transduction pathways that have the ability to modify a variety of cellular functions, including cardiac remodeling, cell survival, apoptosis, maladaptive hypertrophy, fibrosis, and mitochondrial function. It can also regulate the phosphatidylinositol 3-kinase/protein kinase B (PI3K/Akt) pro-survival pathway by activating SIRT1. Enhanced Akt activation can inhibit ROS-mediated apoptosis and improve cardiomyocyte survival through upregulation of FOXO transcription factors, inhibition of the Bcl-2 transcription factor family as well as glycogen synthase kinase-3β (GSK-3β) [109,110]. In addition, RES suppresses the ROS-mediated activation of MAPKs, which are important factors in controlling several cellular processes such as proliferation and apoptosis under both normal and pathological conditions. Overexpression of MAPKs induced by ROS has been shown to be involved in pathological cardiac hypertrophy and remodeling [111]. RES can stimulate the production of MAPK phosphatase-1 (MKP-1), which is the major negative regulator of MAPK activity [112,113]. However, it can inhibit cardiac remodeling through other pathways as well. RES can reduce Ang II-induced cardiac hypertrophy by inhibiting adverse NF-κB signaling activation [114]. Moreover, the downregulation of the serine/threonine-specific protein kinase mammalian target of rapamycin (mTOR) and activation of SIRT1 by RES may be important in inhibiting cardiac and endothelial hypertrophy. mTOR is a downstream target of many signaling pathways including AMPK and PI3K/Akt and has a central role in the regulation of cell growth, survival, and apoptosis, as well as transcription [115,116,117]. Besides maladaptive hypertrophy, cardiac fibrosis is also a key feature of remodeling defined by increased collagen secretion and deposition in the extracellular matrix (ECM) by excessive fibroblast activation [118]. Transforming growth factor β (TGF-β)/Smad2/3 signaling pathway plays a central role in the regulation of ECM production; however, RES-mediated SIRT1 activation can alleviate fibroblast activity by downregulating the TGF-β/Smad2/3 signaling pathway via overexpressing the Smad7 inhibitor protein and silencing miR-17 gene [119,120]. Moreover, RES inhibits phosphatase and tensin homolog (PTEN) degradation, which is a negative regulator for a large number of signaling pathways (PI3K/AKT/mTOR and TGF-β/Smad2/3) and a positive regulator of AMPK (AMP-activated protein kinase). Therefore, it can favorably moderate cardiac hypertrophy and fibrosis [116,121].

Endothelial cells regulate vascular tone, vascular resistance, blood pressure, and hemostasis through the release of vasoactive substances, such as PGs, NO, and vasoconstrictive endothelin-1 (ET-1). Therefore, they play an essential and dynamic role in regulating the CV system. [122,123]. Normal vascular homeostasis can be deranged by various noxious insults and under different pathologic conditions, such as chronic inflammation and oxidative stress. The abnormal upregulation of ET-1 and the reduction in NO levels were shown to be associated with atherosclerosis development and CVDs. Besides their anti-inflammatory and antioxidant properties, stilbenes can modulate the activity of several signaling pathways and improve endothelial function. The beneficial effects of stilbenes on endothelial function and vascular wall tension have been previously linked to increased NO production and decreased ET-1 synthesis. RES enhances NO synthesis in endothelial cells by upregulating eNOS expression and preventing eNOS uncoupling through multiple mechanisms, such as SIRT1 activation [124,125,126]. Furthermore, RES can inhibit the overproduction of ET-1 induced by RAAS and various pro-inflammatory cytokines [127,128]. Piceatannol can also protect endothelial function by enhancing eNOS phosphorylation and increasing NO production [106].

Mitochondria serve as a crucial source of energy for cells as they produce adenosine triphosphate (ATP) in the process known as oxidative phosphorylation [129]. The beneficial effects of RES on mitochondrial biogenesis and function are thoroughly investigated. The activation of the AMPK/SIRT1/PGC1α pathway is a major axis related to mitochondrial biogenesis and the regulation of cellular energy metabolism. AMPK is an energy sensor that controls cellular metabolic function and balances ATP generation. It is stimulated by an increased AMP-to-ATP ratio and in response, shifts cell metabolism towards ATP generation [130]. Downregulation of NOX isoforms by RES induces phosphorylation (stimulation) of AMPK and increases the level of nicotinamide adenine dinucleotide (NAD+), thereby inducing SIRT1 expression (NAD+-dependent protein deacetylase). SIRT1 stimulates deacetylation and activation of the transcriptional coactivator PGC-1α (peroxisome proliferator-activated receptor gamma coactivator 1-alpha), thereby enhancing Nrf-1 and Nrf-2 transcription factors, which regulate the expression of genes encoding certain proteins of the electron transport chain (ETC) [131,132,133]. Moreover, experimental trials demonstrated that RES effectively regulates mitochondrial dynamics (fusion and fission) and other quality control processes in cardiomyocytes through activation of the Sirt1/Sirt3/PGC-1α signaling pathways and enhancing the expression of Drp1 (a regulator protein of mitochondrial fission), among others [134,135].

### 3.2. Pre-Clinical and Human Clinical Trials with RES Related to CVDs

RES is found at high concentrations in grape skin thus red wine represents an important dietary source (Figure 1). A large number of pre-clinical and human clinical trials have published on the beneficial effects of this stilbene in CVDs and we aim to provide a brief review of pertinent data below (Table 1). Overall, RES is well-tolerated over a wide daily dose range (5 mg–5 g) with mild side effects described in human clinical trials when exceeding 500 mg/day intake. However, the exact cut-off dose for toxicity remains unknown to date.

#### 3.2.1. Pre-Clinical Trials

##### Vascular Function, Hypertension

Based on several pre-clinical studies, RES has a potent vasorelaxant and thus antihypertensive effect. Multiple mechanisms contribute to this favorable property, including SIRT1 activation, AMPK phosphorylation, increased eNOS activity, and reduced ROS production [185,186,187,188]. Per Dolinsky et al., 320 mg/kg daily RES intake for 5 weeks significantly reduced the blood pressure in spontaneously hypertensive rats (SHRs) as well as mice receiving angiotensin II (AT-II) infusion. Improved vascular relaxation by enhanced eNOS activity and reduced oxidative stress contributed to this finding [189]. Grujic-Milanovic et al. further investigated the antihypertensive and antioxidant effects of RES in rats [190]. SHRs were fed RES at a daily dose of 10 mg/kg or placebo for 8 weeks, which led to an improvement in both systolic (172.04 ± 9.68 vs. 138.21 ± 16.05 mmHg; *p* < 0.01) and diastolic blood pressure (133.33 ± 12.65 vs. 110.85 ± 14.07; *p* < 0.01). Redox status and renal function also showed significant improvements in the RES group [190]. Similar results were found by Franco et al. in obese adult rats receiving RES supplementation at 30 mg/kg/day for 30 days [191]. In addition to the above experimental designs, RES was shown to effectively reduce blood pressure in other animal models of hypertension, including renal artery clipping (RES: 5, 10 or 20 mg/kg/day for 4 weeks) [192], hypoxia-induced (RES: 10 mg/day for 4 weeks), AT-II infusion (RES: 146 mg/day for 4 weeks) [185], and rodents fed with fructose [187,193]. Interestingly, Prysyazhna et al. described a paradox mechanism of action for RES in hypertensive mice. While RES reduced blood pressure in their study, the vasorelaxation was caused by cGMP-dependent protein kinase 1α (PKG1α) activation via direct protein oxidation. This pro-oxidant effect is likely due to free radical formation upon electron transfer to oxygen, especially in ROS-damaged tissues, thus facilitating favorable signaling directly at the site of injury [194]. The protective effect of RES on vascular function was further investigated in numerous in vivo and in vitro experimental studies [195,196,197,198].

##### Cardiac Function, Remodeling

It is well established that oxidative damage and chronic inflammation stimulate specific intracellular signal transduction pathways resulting in adverse cardiac remodeling characterized by fibrosis, cardiomyocyte hypertrophy, apoptosis, and necrosis [199]. Given its potent antioxidant and anti-inflammatory properties, RES was extensively evaluated in different murine models of cardiac remodeling and HF. Gu et al. studied the efficacy of RES at a dose of 2.5 mg/kg/day administered for 16 weeks in a mouse model of HF caused by myocardial infarction (surgical left coronary artery ligation). The results showed improved cardiac function and survival in the group receiving RES, potentially due to SIRT1-mediated upregulation of the AMPK signaling pathway [200]. These results are in agreement with another post-infarction, pre-clinical trial performed by Ahmed et al. [201]. It this study, long-term (10-month) RES treatment at a daily dose of 5 mg/kg significantly improved left ventricular (LV) systolic function in rats [201]. Similar results were published by Kanamori and colleagues [202]. In a subsequent study by Riba et al., RES administered at 15 mg/kg/day for two months reduced the severity of myocardial dysfunction and improved LV function in an isoproterenol-induced post-infarction HF model. These beneficial effects of RES were attributed to its ability to modify the activity of various intracellular signaling pathways (Akt-1/GSK-3β, p38-MAPK, ERK1/2, and iNOS uncoupling) [28]. In a hydroxyeicosatetraenoic acid (HETE)-induced post-infarction rat model of HF, RES supplementation at 5.82 mg/kg/day significantly increased systolic LV function and successfully reversed LV and left atrial (LA) remodeling [203]. Wojciechowski et al. demonstrated in a pressure overload-induced myocardial hypertrophy model that RES can alleviate the progression of cardiac remodeling, dysfunction, and reduce oxidative injury (lipid peroxidation) in cardiac tissue even at lower doses (2.5 mg/kg/day) [204]. In a study by Rimbaud et al., RES supplementation at 18 mg/kg/day for 2 months was associated with decreased cardiac remodeling and improved cardiac function in a high salt diet-induced hypertensive rat model of HF. In addition, RES increased the antioxidant activity of cardiomyocytes, improved mitochondrial function by enhancing PGC-1α cascade activation, and decreased mitochondrial lipid oxidation [205]. Conversely, Sung et al. published that RES supplied at a higher dose (150 mg/kg/day) did not significantly change systolic LV function after 2 weeks, but improved diastolic function, reduced LV diameters/volumes, and was able to reverse cardiac remodeling (fibrosis, hypertrophy) [206]. In another set of experiments, RES-induced SIRT1 activation was shown to represent a key protective mechanism against cardiac dysfunction. Ma et al. have shown that RES supplemented at 25 mg/kg/day inhibits myocardial hypertrophy and fibrosis in mice with HF by enhancing SIRT1 activity, thereby upregulating the PGC-1α/Nrf-1 and Nrf-2 pathways and improving mitochondrial function [133]. The experiments by Bagul et al. have provided further evidence supporting the antihypertrophic effects of RES mediated by SIRT1 when it was used at 10 mg/kg/day for 8 weeks in diabetic rats [84]. Considering outcomes beyond cardiac function, RES given at 450 mg/kg/day for 2 weeks significantly improved exercise capacity and reduced fatigue by improving flow-mediated vasodilatation and vascular function in a mouse model of HF induced by pressure overload [207]. Further pre-clinical trials suggested that RES may pre-condition the myocardium against ischemia-reperfusion (I/R) damage by activating antioxidant, anti-inflammatory, pro-survival, and anti-apoptotic pathways [208,209,210]. However, this benefit seems to be highly dependent on the dose and duration of administration [211].

In addition to the studies detailed above, many others have examined the potential beneficial effects of RES on HF provoked by pressure overload [204,212], diabetes mellitus [213], chemotherapy [214], sepsis [215], myocarditis [216], and genetic mutations [217].

##### Other Conditions, Risk Factors

Its anti-atherogenic effects are based primarily on the ability to improve vascular and endothelial function as well as favorably modifying the CV risk factor profile. RES supplementation has been shown to reduce serum lipid levels (total cholesterol, LDL-C, and triglyceride) [218,219,220], regulate glucose homeostasis [221,222], reduce vascular inflammation and oxidative stress [223,224] through the downregulation of NF-κB and p38-MAPK pathways and upregulating SIRT1 expression [225,226].

#### 3.2.2. Human Clinical Trials

##### Hypertension

The success of RES in pre-clinical studies prompted a range of human clinical trials involving populations with various diseases. Theodotou et al. investigated the efficacy of RES supplementation in patients with essential hypertension, adding it on top of standard therapy. A total of 97 patients were enrolled, divided into two groups, and followed for 2 years: (1) RES (50 mg/day) + standard treatment; (2) standard of care alone. The long-term administration of RES led to a reduction in blood pressure when initiated on top of standard medical therapy [145]. Conversely, Walker et al. were unable to detect a significant biological effect of RES given at 1 g/day for 35 days on blood pressure in patients with metabolic syndrome [146]. Similar results were published by Gal et al. in patients with systolic HF [66]. Importantly, these two studies enrolled normotensive patients suggesting that RES supplementation does not provoke hypotension making it a potentially safe agent in those with low baseline blood pressure (such as advanced systolic HF). In a recent meta-analysis by Fogacci et al., low-dose RES did not significantly affect either systolic or diastolic blood pressure but administering it at a higher dose (>300 mg/day) reduced both values significantly [140]. In another meta-analysis, Liu et al. found that daily RES consumption exceeding 150 mg effectively reduced systolic (−11.9 mmHg, *p* = 0.01) but not diastolic blood pressure [141]. Several individual trials provided data regarding the antihypertensive effect of RES in the setting of different comorbidities, including diabetes [142,143], obesity [144], and fatty liver disease [227].

##### Vascular Protection

Although the vascular protective effects of RES are well-established in pre-clinical experiments, strong human evidence is lacking. In a pilot trial by Wong et al., RES given at 270 mg/day for 4 weeks led to a significant improvement in endothelial function as measured by flow-mediated dilatation (FMD) of the brachial artery in overweight and hypertensive individuals [147]. Similarly, in a study published by Fujitaka et al., RES (100 mg/day for 3 months) improved endothelial function as measured by FMD in 34 patients with metabolic syndrome [228]. However, no significant change was noted in blood pressure [228]. A study by Magyar et al. found that, compared to placebo, low-dose RES intake (10 mg/day given for 3 months) improved vasorelaxation significantly in a population with chronic angina pectoris [27]. While these studies established the beneficial effects of long-term RES supplementation, a single dose may also be effective. Marques et al. showed that 300 mg of RES improved endothelial function as measured by FMD but there was no change in peripheral and central (aortic) systolic blood pressure [148]. In addition to its beneficial effects on the vasculature, there is ample evidence regarding the potent anti-atherogenic and CV risk-modifying properties of RES, including lipid profile, diabetes, and inflammation [27,69,136,137,138,139,229]. Altogether, these data suggest that RES may be considered an effective compound for CV prevention.

##### Heart Failure

In contrast to the high number of pre-clinical studies with RES in various HF models, merely a few human clinical trials have been performed to date in populations with HF. In a double-blind, placebo-controlled, randomized clinical trial, Magyar et al. investigated the efficacy of RES intake in post-MI patients with Stage B HF and preserved ejection fraction (baseline EF: 54.77 ± 1.64% in the treated group) [27]. Not unexpectedly, 10 mg/day RES for 3 months did not significantly impact the ejection fraction. However, it significantly improved the diastolic function parameters compared to placebo [27]. In another randomized clinical trial by Militaru et al. involving patients with chronic stable angina pectoris, RES supplemented at 20 mg/day for 2 months led to a significant reduction in serum N-terminal prohormone brain natriuretic peptide (NT-proBNP) levels [230]. To date, only one double-blind, placebo-controlled, randomized clinical trial evaluated the potential benefits of RES in patients with symptomatic systolic HF [66]. The study was performed by our group at the University of Pécs. Sixty patients with New York Heart Association (NYHA) class II-III systolic HF were enrolled and randomly assigned to one of two cohorts: (1) 100 mg/day RES supplementation for 3 months; (2) placebo. All subjects received maximally tolerated guideline-directed medical therapy for HF at the time of enrollment. At the end of the study period, systolic and diastolic function as well as global longitudinal strain improved in the RES cohort with significantly lower serum biomarker (NT-proBNP and galectin-3) and inflammatory cytokine (IL-1 and IL-6) levels. Transcriptomic data indicated attenuated leukocyte activation suggesting that RES has a significant anti-inflammatory effect that may serve as a crucial mechanism mediating its benefits in HF. Important from the patient’s perspective, exercise capacity, as well as the reported quality of life (Qol) were higher in the treatment group [66].

## 4. Effect of Relevant Flavonoid Compounds on CVDs

Flavonoids are a large group of polyphenolic compounds found in fruits, vegetables, nuts, seeds, coffee, wine, and tea. They are strong antioxidants that are responsible for the color of fruits and flowers, protect them against biotic stressors, and act as preservatives. Several pre-clinical and clinical studies have demonstrated the beneficial cardioprotective effects of dietary flavonoid intake. The most relevant flavonoids with regard to CVD prevention include flavonols (e.g., Qct, kaempferol), flavanones (e.g., hesperidin), flavanols (e.g., catechin, epicatechin), and anthocyanidins (e.g., malvidin) [231,232].

### 4.1. Mechanism of Action

Flavonoids have several, extensively investigated favorable biochemical effects such as anti-inflammatory, anti-aging, and anti-cancer properties. However, the primary biological activity of these compounds is their profound antioxidant capacity exerted via multiple mechanisms: (1) elimination of ROS; (2) prevention of ROS and RNS production; (3) activation of the antioxidant systems [233,234]. Flavonoids exhibit a marked scavenger activity. The functional hydroxyl group attached to the central ring can donate an electron to peroxynitrite, hydroxyl, and peroxyl radicals through resonance, stabilize these, and create a relatively stable flavonoid radical [235,236,237]. Furthermore, certain flavonoids can directly inhibit the activity of ROS-generating enzymes such as xanthine oxidase, NOS, and NADPH oxidases (NOX isoforms) [238,239,240]. They can also increase the activity of the antioxidant defense system through functional upregulation of selected enzymes such as NADPH-quinone oxidoreductase, glutathione S-transferase, and UDP-glucuronyl transferase [241,242]. In a recent human trial, Zhang et al. demonstrated that anthocyanin significantly increased SOD activity over baseline after 6 weeks of supplementation (*p* < 0.05), and in a dose-dependent manner [243]. Finally, the antioxidant potential of flavonoids may be further enhanced by modification of the chemical structure, such as polymerization (hydroxyl groups) [244], or glycosylation of anthocyanidins [245].

Several trials reported that flavonoids have potent anti-inflammatory properties. They inhibit prostaglandin synthesis, thereby decreasing leukocyte infiltration and edema development [246]. Some block arachidonic acid release through the inhibition of phospholipase A2 (PLA_2_) as well as phospholipase C1 (PLC_1_). Moreover, Qct can block the synthesis of prostaglandins, leukotrienes, and thromboxanes by inhibiting the cyclooxygenase (COX) and lipoxygenase enzymes [247,248,249]. Many flavonoids (e.g., catechin, Qct) can decrease pro-inflammatory cytokine expression, such as IL-6, IL-8, TNF-α, and IL-1β, in macrophages and T-cells while enhancing the production of anti-inflammatory cytokines (e.g., IL-10) [247,250,251]. Nam et al. demonstrated in mice that rutin, a Qct derivative, is not only able to reduce pro-inflammatory cytokine levels but can also decrease NO production by inhibiting iNOS and COX-2 expression [252]. In a recent human trial, Zhang et al. found that anthocyanin supplementation significantly decreased pro-inflammatory cytokine levels, a favorable effect that was dose-dependent [243]. In addition, apigenin successfully inhibited the TNF-α-induced intercellular adhesion molecule-1 (ICAM-1), E-selectin, and vascular cell adhesion molecule-1 (VCAM-1) expression on the surface of endothelial cells [247,251].

Flavonoids have the ability to modulate numerous intracellular signaling factors and signal transduction pathways, such as NF-κB, AP-1, Nrf2, MAPKs, ERK, PI3K/Akt, to reduce oxidative stress and inflammation, thereby acting as cardioprotective agents [253,254,255,256]. NF-κB regulates the expression of several inflammation-associated genes. Inhibition of this central transcription factor can therefore be of significant benefit. Flavonoids inhibit the nuclear translocation of p50 and p65 subunits of NF-κB in macrophages thus decreasing the expression of pro-inflammatory cytokines, adhesion molecules, NOS, and COX-2 [257,258]. In addition, the flavonoid-induced blockade of IκB-α protein phosphorylation can lead to NF-κB pathway inactivation, thereby suppressing TNF-α-induced apoptosis and inflammation [259,260]. Many flavonoid compounds can block pro-inflammatory cytokine production by modulating the MAPK pathway at different levels. In some pre-clinical trials, the inhibition of MAPK phosphorylation (ERK1/2, p38-MAPK, JNK) by flavonoids prompted a decrease in the transcription and expression of pro-inflammatory cytokines, such as TNF-α [261,262,263]. On the other hand, Qct and rutin-provoked downregulation of MAPK phosphorylation led to protection against Ang II-induced hypertrophy in cardiomyocytes [264]. Moreover, flavonoid compounds have significant anti-fibrotic effects owing to the downregulation of TGF-β/SMADs signaling pathways [265,266]. In addition, they can alleviate oxidative stress-induced damage by targeting the PI3K-AKT–mTOR pathway and inhibiting eNOS uncoupling [267,268].

The protective effect of flavonoids on mitochondria is also extensively studied. They can improve mitochondrial function, and cellular energy metabolism, decrease mitochondrial ROS production, and upregulate the expression of selected genes of the respiratory chain complex I (e.g., heart-specific Ndufs4) [269,270]. In addition, they can modulate the process of mitochondrial dynamics (fission and fusion) by inhibiting OMA1 activity [271].

Finally, flavonoids can improve endothelial function and have potent vasorelaxant properties owing to their ability to suppress endothelin-1 activity and increase NO production. Many flavonoids (e.g., Qct, kaempferol, luteolin) can activate the cAMP/protein kinase A (AMPK) signaling pathway, which further upregulates eNOS, resulting in increased concentrations of endothelial NO [272,273,274]. In addition, flavonoids lower intracellular calcium ion concentration by blocking voltage-gated calcium channels [275].

### 4.2. Pre-Clinical and Human Clinical Trials with Selected Flavonoids Related to CVDs

#### 4.2.1. Quercetin

One of the most widely studied flavonoids is quercetin (Qct) and its derivatives (Qct-glycosides, Qct-ethers, Qct-prenyls) found in apple, blueberry, onion, pepper, broccoli, and tea. They have potent anti-inflammatory and antioxidant effects that are primarily responsible for their positive CV effect (Table 1).

##### Pre-Clinical Trials

Cardiac function, remodeling

The cardioprotective ability of Qct has been broadly examined using in vivo experimental models. Arumugam et al. demonstrated that Qct dosed at 10 mg/kg can protect against the progression of experimental autoimmune myocarditis to dilated cardiomyopathy in rats by inhibiting oxidative stress through modulation of the endothelin 1/MAPK signaling pathways [276]. In another rat model, Qct (25 mg/kg/day) decreased myocardial remodeling and fibrosis by reducing mitochondrial ROS production and total Ca^2+^ content [277]. According to Kumar et al., pre-treatment with Qct (50 mg/kg/day) for 14 days in rats significantly attenuated the isoproterenol-induced acute myocardial injury and inflammation [278]. Putakala et al. showed that Qct (200 mg/kg/day) protects from cardiac oxidative damage caused by high-fructose intake [279]. In addition, it can protect against myocardial I/R injury. According to Liu et al., it decreases I/R-induced oxidative damage and cardiomyocyte apoptosis in mice through NF-κB pathway inhibition and by PPARγ activation [280]. Based on several studies, Qct can effectively reduce cardiac hypertrophy, fibrosis, and remodeling, and can protect against the development of HF [281,282,283]. In a recent trial, Chang et al. also investigated the effects of Qct on cardiac function and found that if given daily at 50 mg/kg dosage for 4 weeks, it can improve cardiac systolic function, attenuate cardiac hypertrophy, and fibrosis in rats by modulating the TGF-β signaling pathway. Moreover, Qct decreased inflammatory cytokine levels (TNF-α, IL-13, IL-18) and ROS production by improving mitochondrial function and activating SIRT5 expression [284].

Vascular function

The vascular endothelial benefits of Qct are also well demonstrated in pre-clinical trials. Machha et al. found in diabetic rats that the daily intake of 10 mg/kg protected vascular function via antioxidant activity and increased endothelium-derived NO levels [285]. According to a pre-clinical trial by Sánchez et al., Qct administration (10 mg/kg for 13 weeks) improved blood pressure and endothelial function in spontaneously hypertensive rats via enhancing NO production and suppressing the NADPH oxidase-mediated ROS generation [286]. Several researchers reported on the renin–angiotensin–aldosterone system inhibitory potential of Qct in pre-clinical trials that may be responsible, at least in part, for its antihypertensive properties and ability to improve endothelial dysfunction [272,287,288]. Finally, Qct and its derivatives can influence major CV risk factors as well, including diabetes mellitus [289,290,291] and dyslipidemia [292,293].

##### Human Clinical Trials

Hypertension

A limited number of human clinical trials evaluated the beneficial effects of Qct on hypertension. Edwards et al. were the first to investigate the potential benefits of this compound in humans. In a randomized, double-blind, placebo-controlled, crossover study, 19 subjects with pre-hypertension and 22 with stage 1 disease were enrolled and administered 730 mg Qct or placebo daily for 4 weeks. There was no significant blood pressure change in the pre-hypertension group, but a significant reduction was documented in those with stage 1 hypertension: systolic (−7 ± 2 mmHg) and diastolic (−5 ± 2 mmHg) [149]. In another randomized clinical trial by Zahedi et al., 500 mg/day Qct administration for 10 weeks significantly decreased systolic blood pressure compared to placebo (−8.8 ± 9.3 vs. −3.5 ± 11.7 mmHg, *p* = 0.04) in women with type 2 diabetes mellitus (*n* = 72). However, it had no effect on diastolic blood pressure or on any of the other measured variables, such as lipid profile and inflammatory markers [150]. Furthermore, in 2015 Brüll et al. demonstrated the benefits of Qct in a double-blind, placebo-controlled, crossover trial. After randomization, participants (*n* = 70) received 162 mg Qct daily (onion skin extract powder) or placebo for 6 weeks [151]. Systolic blood pressure in hypertensives was significantly lower in the treatment cohort vs. controls receiving placebo. In contrast, using a similar dosing scheme, Dower et al. did not find Qct to significantly reduce blood pressure at 4 weeks (*n* = 37) [152]. However, measurements were made using different methods (office or 24-h ambulatory blood pressure monitoring).

The effect of Qct may be dose-dependent. A meta-analysis by Serban et al. found that supplementation of 500 mg/day and above may significantly reduce both systolic (−4.45 mmHg, *p* = 0.007) and diastolic (−2.98 mmHg, *p* < 0.001) blood pressure values; however, the effect was not significant at lower doses [153]. Evaluating its effect on end-organ function, Kondratiuk et al. found that the addition of Qct to standard therapy in hypertensive patients (*n* = 84) for 12 months significantly improved diastolic echocardiographic parameters when compared to placebo (E/e’: −0.41 ± 0.01 vs. −0.08 ± 0.01; *p* = 0.001; LV mass index: −3.28 ± 0.02 vs. −2.04 ± 0.03 g/m^2^, *p* = 0.02) [157]. In a small double-blind, placebo-controlled trial, Larson et al. investigated the acute effects of Qct in humans. They enrolled 12 subjects with stage 1 hypertension and administered 1095 mg Qct in a single dose that led to a significant reduction in both systolic and diastolic blood pressure at 10 h [154]. When given at a lower dose (50 to 400 mg) as a single dose to healthy individuals, Bondonno et al. found no acute change in blood pressure or in NO-mediated endothelium-dependent vasodilation [155]. Based on these data, the efficacy of Qct on blood pressure may be dose-dependent with longer-term administration needed to observe a significant change.

Myocardial ischemia

The number of studies evaluating the effect of Qct in HF and CAD is limited. Chekalina et al. investigated the potential benefits of Qct on myocardial ischemia and hemodynamic parameters in a population with stable CAD receiving standard-of-care therapy in a randomized, placebo-controlled trial. A total of 85 subjects were randomized into two groups: (1) 120 mg daily Qct supplementation for 2 months; (2) continued standard treatment for CAD. LVEF increased by 4.5% and 3.2%, respectively on echocardiography which also revealed an improvement in diastolic function parameters in both groups except for mitral inflow deceleration time (DT), which only changed in the cohort receiving Qct. In addition, 24-h Holter ECG monitoring revealed a significant reduction in the number of ST segment depression episodes as well as the incidence of premature ventricular complexes in the treatment group [156].

#### 4.2.2. Catechins

The potential CV benefits of flavanols (catechin, epicatechin, and their derivative epigallocatechin) have been extensively studied in the past decades (Table 1). The major sources of catechins are apples, pears, cocoa beans, tea, and grapes. Therefore, it is present in popular consumer products such as chocolate, tea (especially green tea), and wine. It is important to point out that flavanols are the major biologically active polyphenol components of green tea and cocoa, therefore many studies have been published using extracts from these products.

##### Pre-Clinical Trials

Vascular function, hypertension

The antihypertensive and vascular protective effects of catechins are well demonstrated in pre-clinical trials. Catechins were shown to decrease blood pressure, improve endothelial function, and induce vasorelaxation [273,274,294,295]. In a recent trial, Munoz et al. investigated the beneficial effects of green tea extract on the CV system in animal models. Supplementation for 20 weeks prevented the development of hypertension and improved endothelial function in obese mice by decreasing arterial inflammation and oxidative stress [296]. In a similar study, Gao et al. found that green tea consumption can decrease salt-induced hypertension in elderly rats by inhibiting the renin–angiotensin–aldosterone system, upregulating eNOS synthesis, improving antioxidant activity, and downregulating pro-inflammatory processes (IL-1β) [297]. In addition, Sabri et al. demonstrated that daily epigallocatechin gallate (EGCG) supplementation at 50 mg/kg for 14 days decreases blood pressure and alleviates vascular dysfunction caused by increasing NO bioavailability in hypertensive mice receiving angiotensin II infusion [298].

Several animal studies performed in the past decades have demonstrated the beneficial effect of catechins on serum lipid levels and their ability to reduce atherosclerotic plaque development by inhibiting pro-inflammatory cytokine formation (e.g., TNF-α) and oxidative stress-induced production of oxidized LDL [299,300,301,302,303].

Cardiac function, remodeling

Ischemia-induced oxidative damage of the myocardium leads to cardiomyocyte hypertrophy, myocardial fibrosis, and apoptosis. The myocardial protective effects of catechins during I/R insults may be explained by their antioxidant and anti-inflammatory properties. Ferenczyová et al. provided a thorough summary of the most relevant results related to I/R injury [304]. It is important to emphasize that catechin doses used in pre-clinical studies varied broadly (1–200 mg/kg/day).

Mou et al. published an experimental trial in 2022 using chronic pressure overload to induce HF [270]. Following abdominal aortic constriction (AAC) surgery, rats were fed EGCG for 8 weeks. Authors found that EGCG supplementation helped preserve cardiac function and inhibited myocardial hypertrophy/fibrosis by downregulating the TGF-β signaling pathway activity. In addition, it improved cellular energy metabolism by improving mitochondrial function [270]. Muhammad et al. published similar results with EGCG supplementation at 200 mg/kg/day for one month in aged Wistar albino rats [305]. In addition to these papers, the anti-hypertrophic, anti-fibrotic, and anti-apoptotic properties of catechins have been well described in other pre-clinical animal studies [306,307,308].

##### Human Clinical Trials

Hypertension

Several human clinical trials have investigated the vasoactive and anti-hypertensive activity of catechins [309,310,311]. Bogdanski et al. evaluated the effect of green tea extract on cardiovascular risk factors in obese, hypertensive individuals. In this double-blind, placebo-controlled study, patients (*n* = 56) received green tea extract (379 mg/day) or placebo for 3 months. Both systolic (−4.9 ± 5.7 mmHg) and diastolic (−4.7 ± 3.2 mmHg) blood pressure values were significantly lower in the treatment group compared to placebo at the end of the follow-up period [158].

Rostami et al. provided further evidence on the ability of catechin-rich cocoa products to reduce blood pressure in patients at high CV risk [159]. In addition, 150 mg/day EGCG supplementation effectively lowered blood pressure in obese patients as published by Chatree et al. [160]. Contrary to the above trials, a study by Dower et al. failed to show a significant change in blood pressure (in-office and 24-hour ambulatory monitoring) and vascular function with 100 mg/day epicatechin supplementation for one month [152]. In a meta-analysis, Ellinger et al. found that epicatechin successfully reduced both systolic and diastolic blood pressure values, but the effect was dose-dependent, and the daily intake had to exceed 25 mg [161]. The positive vascular effects of catechins were also confirmed in smokers. Oyama et al. found that high dose catechin intake (580 mg/day) for 2 weeks improved endothelium-dependent vasorelaxation by increasing NO levels and decreasing ROS activity in this population [162]. The dose-dependent vasodilator activity of catechins was evaluated in a short, double-blind, placebo-controlled trial. Twenty healthy males were enrolled and received different doses of epicatechin (0.1, 0.5, and 1.0 mg/kg) or water as control. There was a significant improvement in vascular function as measured by FMD one hour after 1 mg/kg epicatechin intake and two hours after 0.5 mg/kg when compared to placebo [163].

Atherosclerosis and myocardial ischemia

Catechins can also favorably influence the progression of atherosclerosis through CV risk factor modification, such as diabetes and lipid metabolism [152,160,164,312]. It is well known that elevated oxLDL levels induced by oxidative stress are primarily responsible for the development and progression of atherosclerosis and in general, CVDs. In a randomized, placebo-controlled, double-blind trial, Suzuki-Sugihara examined the antioxidant capacity of catechins in healthy male volunteers with 19 subjects eating a total of 1 g of catechin each. Plasma total antioxidant capacity was elevated at one hour following ingestion and LDL oxidizability was significantly reduced [165]. In a small, phase II, double-blind, randomized clinical trial (COCOA-PAD study), patients with known PAD consumed 75 mg epicatechin daily in the form of cocoa (*n* = 44) or placebo. The average six-minute walking distance increased significantly in the treatment group compared to the placebo at 6-month follow-up (+42.6 m; *p* = 0.005) [166]. In a human study published more recently, Kishimoto et al. examined the association between green tea and CAD. Investigators randomized 612 subjects between 2008 and 2017 and performed coronary angiography identifying 388 patients with CAD and 138 with MI. Individuals were divided into three groups according to the frequency of green tea consumption (1 cup/day, 1–3 cups/day, and 3 or more cups/day). Higher green tea consumption was significantly inversely associated with CAD and MI prevalence in Japanese adults compared to lower green tea intake (OR: 0.54, 95% CI: 0.30, 0.98) [167]. Wang et al. found similar benefits for catechins in a meta-analysis: green tea consumption (one cup daily) was associated with a 10% CAD risk reduction [168].

Heart failure

Data on the effect of catechin supplementation in HF are limited. In a recent trial, Dural et al. examined the benefits of chocolate (containing high amounts of catechins) in patients with systolic HF. They found that NT-proBNP levels decreased significantly with the use of both milk chocolate (356 ± 54.2 vs. 310 ± 72.1 pg/mL; *p* = 0.007) and dark chocolate (341 ± 57 vs. 301 ± 60.1 pg/mL; *p* = 0.028) compared to placebo. However, six-minute walking distance was unaffected [169].

#### 4.2.3. Anthocyanins

Anthocyanins or anthocyanidins (aglycon form) include cyanidin, delphinidin, pelargonidin, petunidin, and malvidin. They are naturally occurring, water-soluble pigments responsible for the color of fruits and vegetables, especially berries (predominantly blueberry, cranberry, and strawberry), grapes, and carrots. The habitual intake of anthocyanins is variable worldwide (12–44 mg/day), but wine consumption contributes to approximately 14.4–24.5% of the total intake in Europe. Anthocyanins became of interest as natural therapeutic polyphenols owing to their potent antioxidant and anti-inflammatory properties with proven efficacy against cardiac and vascular disorders (Table 1) [313].

##### Pre-Clinical Trials

Cardiac function, remodeling

Several pre-clinical studies have recently demonstrated the favorable effect of anthocyanins on myocardial hypertrophy, fibrosis, and remodeling by limiting oxidative stress, reducing inflammation, and modulating cellular energy metabolism. In an in vivo murine model by Chen et al., high-dose delphinidin (15 mg/day for 8 weeks) successfully reversed pressure-overload-induced cardiac hypertrophy, LV dysfunction, and dilation, as provoked by transverse aortic constriction (TAC) in mice. One of the key possible mechanisms for these findings is the attenuation of TAC-induced oxidative stress by delphinidin. Notably, no significant benefit could be detected when it was administered at a lower dose (5 mg/day) [314]. In a similar animal model, Hu et al. used 500 mg/kg daily supplementation of anthocyanin-enriched blueberry extract for 6 weeks in mice. Compared to controls, anthocyanin significantly decreased cardiac mass, myocardial hypertrophy, and fibrosis, and increased LV systolic function by alleviating TAC-induced pro-inflammatory cytokine secretion (IL-1β, TNF-α) and ROS production [315]. Aloud et al. investigated the efficacy of anthocyanins on the cardiac function of SHR. Five-week-old male rats received 10 mg/kg anthocyanin or placebo (water) for 15 weeks. Cardiac mass was lower (less cardiac hypertrophy) and diastolic function was better in the treatment group at the end of the follow-up period. However, blood pressure and LV systolic function were unaffected [316]. The cardioprotective effect of anthocyanins was also demonstrated by Chen et al. in a streptozotocin-induced type I diabetes rat model. Authors showed that 250 mg/kg/day anthocyanin extract for 4 weeks significantly attenuated cardiac hypertrophy, reduced fibrosis, and improved LV function indices [317]. Yue et al. reported using another animal model that anthocyanin reduced myocardial fibrosis and hypertrophy, protected cardiac function, and downregulated the IL-17-related chronic inflammation in diabetic mice [318]. Similar favorable results were found by Liu et al. after cyclophosphamide-induced cardiac injury in rats [319] and by Kim et al. in diabetic mice [320]. Toufektsain and colleagues were the first to provide pre-clinical evidence that prolonged (2 months) anthocyanin intake exerted a myocardial protective effect against ischemia-reperfusion injury in rats through enhancing the activity of the antioxidant defense system [321]. Similar findings were published by Guler et al. in rats [322].

Vascular function, hypertension

The beneficial effects of anthocyanins on vascular function have also been described. Petersen et al. demonstrated that the use of strawberry extract can limit vascular dysfunction by increasing endothelium-dependent vasorelaxation. In addition, it prompts a reduction in blood pressure by inhibiting chronic endothelial inflammation as well as ROS production [323]. Several other authors have published on the anti-inflammatory and vascular protective effects of anthocyanins [324]. For example, Nasri et al. observed that cyanidin-3-glucoside administration at 10 mg/kg for 2 months prompted a significant increase in endothelium-dependent vasodilatation in diabetic rats as compared to placebo [325]. There is further evidence that anthocyanin may lower blood pressure by affecting endothelial vasorelaxation and the activity of the renin–angiotensin–aldosterone system [326,327]. In a pre-clinical trial by Shaughnessy et al., long-term (2 months) blueberry supplementation significantly reduced systolic blood pressure as early as 4 weeks (19% reduction) with continued decline subsequently [328]. As shown by Xu et al., intracranial infusion of anthocyanin extract decreased the sympathetic nervous system activity and reduced blood pressure in salt-induced hypertensive rats [329]. Furthermore, anthocyanins administered at 50, 100, or 200 mg/kg decreased systolic and diastolic blood pressure as well as heart rate in hypertensive rats in a dose-dependent manner, at least in part through RAAS inhibition [330].

Atherosclerosis and risk factors

Several pre-clinical trials have reported on the potent anti-atherogenic properties of anthocyanin-rich fruit extracts, likely related to their lipid-lowering, anti-inflammatory, and antioxidant actions [331,332,333,334,335]. When considering acute CV events, such as MI or stroke, platelet activation (aggregation and adhesion) represents a pivotal event. Therefore, effective inhibition of platelet hyperactivity may be considered a favorable benefit in improving CV outcomes and reducing the risk of adverse events [239]. Importantly, anthocyanins were shown to effectively inhibit collagen- as well as ADP-induced platelet aggregation and adhesion when tested in animal experiments [331,336,337].

##### Human Clinical Trials

Myocardial ischemia

The relationship between anthocyanin supplementation and CVD prevention is relatively well-studied. Cassidy et al. followed 93,600 women (25 to 42 years of age) who were healthy at baseline from the Nurses’ Health Study and examined the association between anthocyanin intake and the risk of MI. During 18 years of follow-up, 405 MI cases were reported and higher anthocyanin intake was inversely associated with the risk of MI after multivariate adjustment (HR: 0.68; 95% CI: 0.49, 0.96; *p* = 0.03 highest vs. lowest quintiles). These data suggest that high habitual anthocyanin intake may effectively reduce the risk of MI in a population of predominantly young women [170]. In a similar study, Cassidy et al. also examined the effect of anthocyanin supplementation on healthy men (*n* = 4046). During the 24 years of follow-up, higher habitual intake of these flavonoids was associated with a reduced non-fatal MI and ischemic stroke risk [171].

In addition to the pre-clinical studies detailed above, the antiplatelet properties of anthocyanins were also evaluated in small human clinical trials. Alvarez-Suarez et al. reported that the daily intake of 500 g of strawberries for one month significantly decreased platelet activation and improved the antioxidant status [172]. Aboonabi et al. were also able to demonstrate the antiplatelet activity of 320 mg daily anthocyanin supplementation for one month in humans [173]. Similar results were reported by Santhakumar and colleagues [174] and Tian et al. published the dose-dependent effect of anthocyanins on platelet function [175].

Hypertension

As potent vasorelaxant and antioxidant agents, anthocyanins can improve vascular function and reduce blood pressure. In a human clinical trial (*n* = 1898), Jennings et al. found that anthocyanin-rich food intake led to a significant reduction in systolic blood pressure and improvement in central hemodynamic parameters, such as augmentation index (AI) and pulse wave velocity (PWV) as measured by non-invasive methods [176]. McKay et al. reported that anthocyanin-rich hibiscus tea consumption for 6 weeks by 65 adults with grade I hypertension reduced systolic blood pressure significantly when compared to placebo (−7.2 ± 11.4 vs. −1.3 ± 10.0 mmHg; *p* = 0.030). However, diastolic blood pressure did not change significantly [177]. Similar benefits were shown by Basu et al. after blueberry consumption [178]. In addition to the aforementioned trials, numerous short-term (less than two months) and long-term studies reported on the antihypertensive effect of an anthocyanin-rich diet (predominantly berries) in hypertensive patients with or without other CV risk factors [179,180,338,339].

Evaluating the acute effects of high anthocyanin intake (480 mL cranberry juice with 94 mg anthocyanin content), Dohadwala et al. found a beneficial effect on vascular function as measured by FMD in subjects with CAD [181]. Several other studies also reported a similar improvement in FMD parameters after short- and long-term consumption of anthocyanin-rich extracts from different berries [340,341,342,343]. In addition to their potent vascular and myocardial protective properties, many trials and meta-analyses were published providing evidence that anthocyanins can positively influence CV risk factors, including diabetes and dyslipidemia [182,183,184,344].

## 5. The Association between CV Mortality and Polyphenol Intake

The number of trials investigating the association between clinical cardiovascular outcomes and polyphenol use is limited. Sesso et al. performed a large, randomized, double-blind, placebo-controlled clinical trial (COcoa Supplement and Multivitamin Outcomes Study; COSMOS) aiming to evaluate the effect of cocoa intake on CVD prevention [345]. Researchers randomized 21,442 adults aged 60 years or older from the US with no prior history of MI or stroke. Participants received daily cocoa extract supplements (500 mg cocoa flavanols, including 80 mg epicatechin) and/or multivitamin supplements or matching placebo. The primary composite outcome was the incidence of stroke, MI, unstable angina requiring hospitalization, need for coronary revascularization, CV mortality, carotid artery surgery, and peripheral artery surgery. After a median follow-up of 3.6 years, 866 participants had a confirmed CV event. Although there was a favorable trend, the study failed its primary composite endpoint (HR: 0.90; 95% CI: 0.78, *p* = 1.02). Analyzing the secondary outcomes, there was a significant reduction in CVD deaths in the group treated with cocoa extract compared to placebo [345]. In another prospective trial, McCullough et al. also reported a mortality benefit with flavonoid supplementation. They randomized almost 100,000 individuals, divided them into quintiles based on their total flavonoid intake, and followed them for 7 years. Comparing the highest quintile group to the lowest, there was a significant reduction in CV mortality (RR: 0.82; 95% CI: 0.73, 0.92; *p* = 0.01). Importantly, there was a signal suggesting that even a relatively small amount of flavonoid-rich food intake may be beneficial for decreasing CV mortality [346]. Similar results were published by Grosso et al. where a high level of flavonoid supplementation was associated with reduced risk of all-cause mortality when compared to low-level consumption (RR: 0.74, 95% CI: 0.55, 0.99, *p* = 0.01) [347]. The PREDIMED trial evaluated the benefits of the Mediterranean Diet on primary CVD prevention with a special focus on the relationship between polyphenol intake and all-cause mortality in patients at high CV risk. Investigators randomized 7447 subjects and assigned them to quintiles based on their total polyphenol intake. During a mean follow-up of 4.8 years, 327 deaths were observed with a 37% relative reduction in all-cause mortality between patients in the highest vs. the lowest quintiles (HR: 0.63; 95% CI: 0.41, 0.97). Subgroup analyses revealed that the mortality benefit was limited to stilbenes (HR: 0.48; 95% CI: 0.25, 0.91; *p*-trend = 0.04) and lignan (HR: 0.60; 95% CI: 0.37, 0.95; *p*-trend = 0.03) [348]. However, more trials are needed to further clarify the potential role of long-term polyphenol intake in CV mortality reduction.

## 6. Red Wine Consumption and Its Association with CVDs

Wine is a complex hydro-alcoholic solution typically derived from fermented grapes. Since ancient times, wine, particularly red wine, has been part of various diets with known health benefits when consumed in moderate amounts [12]. Red wine contains a large variety of bioactive ingredients including polyphenols such as flavonoids (catechin, quercetin, and anthocyanins) as well as stilbenes (RES). These are thought to be responsible for its beneficial effects on the CV system [349]. The polyphenol content of wines depends strongly on the species and the interaction between environmental factors. For example, red wine contains an approximately 10-fold higher concentration of phenolic compounds, especially RES, than white wine with the discrepancy potentially explained by the winemaking method [350].

Renaud et al. were the first to report in 1992 that moderate red wine intake is inversely associated with the prevalence of ischemic heart disease and CV mortality despite the high saturated fat intake in southern France. This phenomenon is known as the “French paradox” [13]. Since then, several human clinical and epidemiological trials have confirmed that regular, long-term red wine consumption in moderate amounts exerts cardioprotective effects and can positively influence CV risk factors [351], lipid profile, and type-2 diabetes [352,353,354,355,356,357]. Our group has shown that moderate red wine consumption at dinner (200 mL for 3 weeks) has a beneficial effect on hemorheological parameters, including red blood cell aggregability and deformability, in healthy volunteers. This may have a positive effect on the microcirculation thereby reducing the risk of CVDs [358]. Naissides et al. reported that short-term red wine consumption had no impact on postprandial lipid parameters or insulin homeostasis, confirming that the benefit is derived from chronic moderate consumption [359].

Gronbaek et al. reported in 2000 that those consuming low (1–7 glasses per week) and moderate (8–21 glasses per week) amounts of red wine have 20% and 24% lower all-cause mortality compared to those not consuming wine, respectively [360]. However, a J-shaped relationship was observed between total alcohol use and mortality. A similar association was described by other investigators [361,362]. Lucerón-Lucas-Torres published a large systematic review and meta-analysis in 2023 focusing on the association between wine consumption and the incidence of CVDs, coronary artery disease, and CV death [363]. Twenty-five clinical studies from 1985 to 2021 were included in the analysis with almost 1.5 million individuals. There was a strong inverse association between red wine consumption and the risk of ischemic heart disease (RR: 0.76; 95% CI: 0.69, 0.84), CVDs (RR: 0.83; 95% CI: 0.70, 0.98), and CV mortality (RR: 0.73; 95% CI: 0.59, 0.90) [363].

The difference between wine and other alcoholic beverages was also analyzed with regard to their CV beneficial effects. For example, Torres et al. demonstrated that red wine consumption increases total antioxidant capacity and improves pro-inflammatory profile, therefore it was more effective in preventing CVD than other alcoholic drinks [364]. It is critical to emphasize, however, that excessive alcohol consumption, including red wine, represents a serious worldwide public health challenge. It is related to the high prevalence (approximately 4% of all adults), its well-established adverse health effects, and social consequences depending on the volume of alcohol intake over time, type of beverage, and consumption pattern. Ethanol exhibits teratogenic, genotoxic, carcinogenic, hepatotoxic, and neurotoxic properties. In addition, it may cause various gastrointestinal adverse effects (gastritis, ulcer formation, bleeding), and can increase the risk of CVD [365,366,367].

In summary, these findings suggest that regular, low-to-moderate intake of polyphenol-rich red wine has a significant beneficial effect in preventing CVDs and reducing CV mortality. However, further research is necessary to better understand the health benefits of red wine consumption, and to define the optimal quantity based on gender and body weight.

## 7. Adverse Effects of the Most Relevant Polyphenols

While polyphenols are well-tolerated compounds overall, it is important to mention a few relevant adverse effects that have been reported in clinical trials.

The side effect profile of stilbenes is the most thoroughly investigated when compared to other polyphenols. The development of adverse effects depends primarily on the duration of administration, achieved plasma concentration, as well as the primary cell/tissue type affected [368,369,370,371,372,373]. RES has been administered over a wide dose range in pre-clinical and human studies, however, the cut-off dose for toxicity remains unknown. The most commonly reported adverse effects were gastrointestinal. Brown et al. found in humans that emesis, diarrhea, and mild hepatic dysfunction may occur at a daily dose of 2.5 g and above [374]. Furthermore, Crowell et al. published that extremely high (3 g/kg/day) oral RES supplementation for 4 weeks resulted in nephrotoxicity (reported as elevated serum creatinine level and renal histopathological changes) in rats [375]. In a human study by la Porte et al., 2 g RES twice daily was overall well tolerated by healthy volunteers, however, diarrhea was observed in six of the eight subjects [376].

Only a few dose-dependent side effects were reported for flavonoids. These generally include gastrointestinal (emesis, diarrhea, abdominal pain, and bloating), in addition to insomnia, headache, and palpitations [377]. Younes et al. published that catechin taken as a dietary supplement in doses exceeding 800 mg/day induces a significant increase in serum transaminases [378]. In another trial, Cruz-Correa et al. reported that the combination of oral curcumin and quercetin for three months only led to a limited number of adverse effects, such as nausea and sour taste [379].

In addition to direct side effects, potentially harmful adverse drug interactions have also been reported with polyphenol consumption. They have been shown to reduce the gastrointestinal transport of folic acid and thiamine and to change the activity of drugs through interaction with drug transporters and digestive enzymes leading to inhibition and increased bioavailability [380]. For example, polyphenols can unfavorably influence iron absorption owing to their chelating effect [381].

## 8. Conclusions and Future Directions

Unhealthy dietary habits are known to contribute to the development and progression of CVDs and mortality. Numerous in vitro and in vivo pre-clinical studies as well as human clinical trials have shown that the consumption of polyphenol-rich diet, including moderate red wine consumption, may be beneficial in the prevention and treatment of CVDs. In this narrative review, we aimed to summarize the primary mechanisms of action for RES and relevant flavonoids in association with CVDs. Their antioxidant and anti-inflammatory properties, and ability to influence various intracellular processes are relatively well described. In pre-clinical trials, RES and flavonoids could effectively reduce blood pressure, enhance vascular function, and attenuate LV hypertrophy and remodeling while improving LV function. In addition, these polyphenols have potent anti-atherogenic effects and can positively influence several CV risk factors such as lipid profile, diabetes, and vascular inflammation. A large number of clinical trials have also been performed confirming the potent cardioprotective effects of these polyphenols in humans.

Despite these published reports, further randomized, interventional clinical studies are needed with extended follow-up time and large cohort enrollment. In addition, it will be critical to establish optimal dosing for each compound and describe potential interaction with other, already approved medications. It is important to emphasize that “natural products or extracts” do not inherently represent ‘safer’ substances than “synthetic products”, therefore detailed investigations are necessary to establish their side effect profile. Based on the literature review, researchers to date have only studied a small fraction of the approximately 10,000 described polyphenols leaving a vast area open for research. Given the increasing prevalence of CVDs, these may prove to be extremely beneficial for cardiovascular prevention.

## Figures and Tables

**Figure 1 biomedicines-11-02888-f001:**
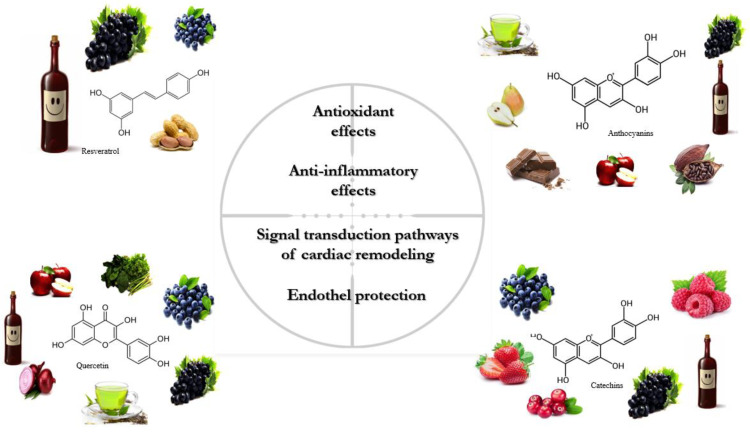
The primary sources and physiological effects of relevant polyphenols.

**Figure 2 biomedicines-11-02888-f002:**
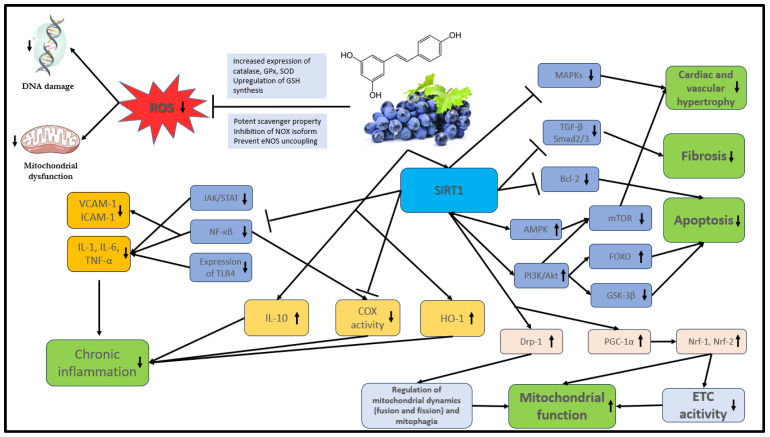
The main mechanisms of action of stilbenes. Akt: protein kinase B; AMPK: AMP-activated protein kinase; COX: cyclooxygenase; Bcl-2: B-cell lymphoma 2, DNA: deoxyribonucleic acid; Drp1: dynamin-related protein 1; ETC: electron transport chain; FOXO: Forkhead box O; GPx: glutathione peroxidase; GSH: glutathione; GSK-3β: glycogen synthase kinase-3 beta; eNOS: endothelial nitric oxide synthase; HO-1: heme oxygenase-1; ICAM: intracellular cell adhesion molecule; IL: interleukin; JAK: Janus kinase; MAPK: mitogen-activated protein kinase; mTOR: mammalian target of rapamycin; NF-κB: nuclear factor kappa B; NOX: NADPH oxidases; Nrf: nuclear respiratory factor; PI3K: phosphatidylinositol 3-kinase; PGC-1α: peroxisome proliferator-activated receptor gamma coactivator 1-alpha; ROS: reactive oxygen species; SIRT1: NAD-dependent deacetylase sirtuin-1; SOD: superoxide dismutase; STAT: signal transducer and activator of transcription proteins; TGF-β: transforming growth factor; TLR4: toll-like receptor 4; TNF: tumor necrosis factor; VCAM: vascular cell adhesion molecule.

**Table 1 biomedicines-11-02888-t001:** The effect of various polyphenols on the cardiovascular system: selected randomized human clinical studies and meta-analyses.

Resveratrol						
Authors	Year	Area of Interest	Study	Study Design	Main Findings	Ref.
Boa et al.	2016	Lipid profile	RCT	120 patients with DM; 500 mg PO daily for 24 weeks	Increased TC levels No change in LDL-C, HDL-C, and TG	[136]
Asgary et al.	2019	Lipid profile	Meta-analysis	396 subjects; 100–3000 mg PO daily	No change in TC levels Increase in HDL-C	[137]
Magyar et al.	2012	Lipid profile, endothelial function	RCT	40 patients with CAD; 10 mg daily for 3 months, oral capsule	Decrease in LDL-C levels only Improvement in FMD	[27]
Hoseini et al.	2019	Metabolic status	RCT	56 patients with CAD + DM; 500 mg PO daily for 4 weeks	Beneficial effects on TC/HDL-C ratio, HDL-C level, glycemic control	[138]
Simental-Mendia et al.	2019	Lipid profile	RCT	71 patients with dyslipidemia; 100 mg PO daily for 2 months	Reduction in TC levels	[139]
Fogacci et al.	2018	Blood pressure	Meta-analysis	681 obese subjects; >300 mg daily	Decrease in SBP	[140]
Liu et al.	2015	Blood pressure	Meta-analysis	247 subjects; >150 mg daily	Decrease in SBP	[141]
Bhatt et al.	2012	Blood pressure, diabetes mellitus	RCT	62 patients with DM; 250 mg PO daily for 3 months	Improved SBP and glycemic control	[142]
Imamura et al.	2017	Blood pressure, arterial stiffness	RCT	50 patients with DM; 100 mg PO daily for 12 weeks	Improved arterial stiffness and blood pressure	[143]
Timmers et al.	2011	Blood pressure	RCT	11 healthy, obese men; 150 mg PO daily for 30 days	Reduction in SBP	[144]
Theodotou et al.	2017	Blood pressure	RCT	97 patients with HT; 50 mg/day PO for 2 years	Reduction in SBP when added to standard therapy	[145]
Walker et al.	2019	Blood pressure, diabetes	RCT	28 men with MS; 2 g PO daily for 30 days	No effect on blood pressure Improved glucose homeostasis	[146]
Wong et al.	2011	Endothelial function	RCT	19 postmenopausal women and obese men with HT; single PO dose (270 mg)	Improved FMD after acute supplementation	[147]
Marques et al.	2018	Endothelial function	RCT	24 subjects with HT; single PO dose (300 mg)	Improved FMD in women after acute supplementation	[148]
Gal et al.	2020	Heart failure, lipid profile	RCT	60 patients with HFrEF; 100 mg PO daily for 3 months	Improved cardiac function, exercise tolerance, QoL, and biomarkers Reduced TC and LDL-C levels	[66]
Quercetin						
Authors	Year	Area of Interest	Study	Study design	Main findings	Ref.
Edwards et al.	2007	Blood pressure	First RCT	41 subjects with HT; 730 mg/day PO for 4 weeks	Reduced SBP and DBP	[149]
Zahedi et al.	2013	Blood pressure, lipid profile	RCT	72 women with diabetes; 500 mg PO daily for 10 weeks	Reduced SBP No effect on lipid profile and DBP	[150]
Brüll et al.	2015	Blood pressure	RCT	70 patients; 162 mg PO daily for 6 weeks	Decreased daytime and nighttime SBP	[151]
Dower et al.	2015	Blood pressure	RCT	37 subjects with HT; 160 mg/day PO for 4 weeks	Reduced BP measured by ABPM	[152]
Serban et al.	2016	Blood pressure	Meta-analysis	587 patients; >500 mg vs. <500 mg/day	Reduced SBP and DBP at doses above 500 mg/day	[153]
Larson et al.	2012	Blood pressure	RCT	12 subjects with HT; 1095 mg in a single dose PO	Acute reduction of SBP and DBP	[154]
Bondonno	2016	Blood pressure	RCT	15 health patients; single dose <400 mg PO	No acute effect on SBP and DBP	[155]
Chekalina	2017	Heart function	RCT	85 subjects with CAD; 120 mg PO daily for 2 months	Improved both systolic and diastolic LV function	[156]
Kondratiuk et al.	2018	Heart function	RCT	84 males with HT; 12 months PO	Improved diastolic function(LV mass and e/e’)	[157]
Catechins						
Authors	Year	Area of interest	Study	Study design	Main findings	Ref.
Bogdansy et al.	2012	Blood pressure	RCT	56 subjects; 379 mg/day PO for 3 months green tea extract	Reduced SBP and DBP	[158]
Rostami et al.	2015	Blood pressure, lipid profile	RCT	60 subjects, high CV risk; 25 g dark chocolate for 8 weeks PO	Reduced SBP and DBP Improved TG level	[159]
Dower et al.	2015	Blood pressure	RCT	37 subjects with HT; 100 mg/day PO epicatechin for 4 weeks	No effect on BP and vascular function	[152]
Chatree et al.	2021	Blood pressure	RCT	30 subjects; 150 mg PO EGCG for 8 weeks	Reduced SBP and DBP Improved TG level	[160]
Ellinger et al.	2012	Blood pressure	Meta-analysis	31 clinical trials; cocoa extracts (epicatechin)	Dose-dependent effect on BP (>25 mg/day epi)	[161]
Oyama et al.	2010	Vascular function	RCT	30 patients; 580 mg/day green tea extract for 2 weeks PO	Improved the endothelium-dependent vasorelaxation	[162]
Alañón et al.	2020	Vascular function	RCT	20 healthy volunteers; different doses of epicatechin	Dose-dependent effect on vascular relaxation (>0.5 mg/kg epi)	[163]
Samavat et al.	2016	Lipid profile	RCT	1075 women; green tea extract (EGCG 843 mg/day) for 12 months PO	EGCG reduced TC and LDL-C level	[164]
Suzuki-Sugihara et al.	2016	Redox status	RCT	19 healthy male subjects; 1 g catechin in a single PO dose	Increased total antioxidant capacity Decreased ox-LDL-C level	[165]
McDermott et al.	2020	PAD	RCT	44 subjects with PAD; 75 mg/day PO epi for 6 months	Improved six-minute walk distance	[166]
Kishimoto et al.	2020	CAD	RCT	612 subjects +/− CAD; green tea extracts PO	Green tea is inversely associated with the prevalence of CAD and MI	[167]
Wang et al.	2011	CAD	Meta-analysis	5 studies on green tea	10% reduction in the risk of developing CAD	[168]
Dural et al.	2022	Heart failure	RCT	20 patients with HFrEF; dark or milk chocolate PO	Reduced NT-proBNP levels Improved vascular function	[169]
Anthocyanins						
Authors	Year	Area of interest	Study	Study design	Main findings	Ref.
Cassidy et al.	2013	Risk of MI	OBS	93,000 healthy subjects; AC-rich diet for 18 years PO	47% reduction in MI risk	[170]
Cassidy et al.	2016	Risk of MI and ischemic stroke	OBS	4046 males; AC-rich diet for 24 years PO	13% reduction in MI risk	[171]
Alvarez-Suarez et al.	2014	Platelet function, lipid profile	RCT	23 healthy volunteers; 500 g daily PO intake of strawberries for 1 month	Decreased platelet activation Improved lipid profile Enhanced antioxidant status	[172]
Aboonabi et al.	2020	CV risk factors, Platelet function	RCT	55 subjects with metabolic syndrome; 320 mg AC-rich extract PO for 4 weeks	Reduced thrombogenicity Improved lipid and glucose homeostasis	[173]
Santhakumar et al.	2015	Platelet function	RCT	21 healthy subjects; AC-rich extract for 28 days PO	Decreased platelet activation	[174]
Tian et al.	2021	Platelet function	RCT	93 subjects with dyslipidemia; various doses for 12 weeks	Dose-dependent attenuation of platelet function (>80 mg/day)	[175]
Jennings et al.	2012	Blood pressure	RCT	1898 subjects; AC-rich PO diet	Improved arterial stiffness and blood pressure	[176]
McKay et al.	2009	Blood pressure	RCT	65 subjects; hibiscus tea PO for 6 weeks	Decreased the SBP	[177]
Basu et al.	2010	Blood pressure	RCT	48 subjects with MS blueberry PO for 8 weeks	Decreased SBP and DBP	[178]
Johnson et al.	2015	Blood pressure	RCT	48 women with HT; 22 g freeze-dried blueberry for 8 weeks PO	Reduced blood pressure and arterial stiffness	[179]
Emamat et al.	2022	Blood pressure	RCT	85 subjects with HT; 10 g/day dried purple-black barberry for 2 months PO	Improved SBP	[180]
Dohadwala et al.	2010	Vascular function	RCT	59 subjects with CAD; 480 mL cranberry juice (94 mg AC) for 4 weeks PO	Improved FMD	[181]
Huang et al.	2016	CV risk factors	Meta-analysis	22 trials with 1251 healthy subjects or with CAD	Reduced LDL-C, glucose levels Improved SBP	[182]
Yang et al.	2017	CV risk factors	RCT	160 participants with DM; 320 mg/day PO for 12 weeks	Improved lipid and glucose profile	[183]
Liu et al.	2016	Lipid profile	Meta-analysis	six trials with 586 subjects with high CV risk	Reduced TC, TG and LDL-C levels Increased HDL-C	[184]

ABPM: ambulatory blood pressure monitoring; AC: anthocyanins; CAD: coronary artery disease; CV: cardiovascular; DBP: diastolic blood pressure; DM: diabetes mellitus; EGCG: epigallocatechin-gallate; EPI: epicatechin; FMD: flow-mediated dilation; HDL-C: high-density lipoprotein cholesterol; HFrEF: heart failure with reduced ejection fraction; HT: hypertension; LDL-C: low-density lipoprotein cholesterol; LV: left ventricle; MI: myocardial infarction; MS: metabolic syndrome; OBS: observational trial; PAD: peripheral arterial disease; PO: by mouth (per os); Qct: quercetin; Qol: quality of life; RCT: randomized controlled trial; RES: resveratrol; SBP: systolic blood pressure; TC: total cholesterol.

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
