# Peer review of "Resveratrol and beyond: The Effect of Natural Polyphenols on the Cardiovascular System: A Narrative Review"

_biomedicines, 2023, doi:10.3390/biomedicines11112888_

Round 1

Reviewer 1 Report

The authors have provided an exhaustively overview of the most relevant natural polyphenols including more than 350 references. They have reviewed their pharmacological properties, mechanisms of action and the status of some of them in pre-clinical and clinical trials. The review topic is the high relevance and it fills the gap in knowledge because it covers subjects which have not  reviewed before.

Author Response

We would like to thank Reviewer#1 for the careful and thorough review of our manuscript and for providing excellent rating. Please see our point-by-point answers to the questions/comments below.

General Comment: The authors have provided an exhaustively overview of the most relevant natural polyphenols including more than 350 references. They have reviewed their pharmacological properties, mechanisms of action and the status of some of them in pre-clinical and clinical trials. The review topic is the high relevance, and it fills the gap in knowledge because it covers subjects which have not reviewed before.

Answer: We sincerely appreciate your kind comments on the relevance and quality of our manuscript. Thank you again for taking the time to review the paper.

Sincerely yours,

László Czopf MD, PhD

Reviewer 2 Report

The authors summarize evidence for the positive outcomes of resveratrol and other related polyphenols on the cardiovascular system. They list prior studies on the classification/biosynthesis/metabolism of polyphenols, stilbenes and CVDs, the effect of Relevant flavonoids on CVDs, the Association between CV Mortality and polyphenol intake, and red wine consumption and CVDs. The authors provided one interesting figure that summarizes the primary sources and physiological effects of relevant polyphenols. Moreover, the authors provided one interesting table that summarizes the effect of various polyphenols on the cardiovascular system: selected randomized human clinical studies and meta-analyses. From this discussion, the authors suggest that several pre-clinical studies and clinical have demonstrated that the consumption of a polyphenol-rich diet, including moderate red wine consumption, may be beneficial in the prevention and treatment of CVDs.

1) The provided sections read like narration for the evidence of discussed points without critical aspects/reflection points. At the end of each section, a take-home message is advised to be provided.

2) In the introduction section, the authors are advised to elaborate on the novelty of the current review and how is it unique/updated relative to the previous reviews regarding the positive outcomes of resveratrol/related flavonoids on the cardiovascular system.

3) In order to attract the interest of more readers regarding the current review, the authors are advised to summarize the narration in section 3.1 (Mechanism of Action) with mechanistic schematic figure(s). 

4) In Table 1, the authors should add the route of administration, adverse events (if any), and retention in trial and other related flavonoids.

5) The authors should consider stratifying the narration on the effect on cardiovascular diseases into subsections of hypertension, atherosclerosis, myocardial infarction, and heart failure.

6) In the current review, the authors are advised to provide potential explanations of the reported controversy among several clinical trials with contrasting outcomes.

7) The authors are advised to add a separate section on the reported adverse effects of resveratrol and other flavonoids, particularly, at higher doses. Moreover, their major interactions with cardiovascular medications should be described.  

8) The “conclusions” section should be replaced by “Conclusions and future directions”.

The work should elaborate on future directions and limitations of previous studies and what is the next step to translate these findings to clinical settings.

9) The authors should describe how they searched the literature, what are the keywords used, how many studies, and how did they select the proper studies (what are the inclusion/exclusion criteria for these studies)?

10) In section 6 (Red Wine Consumption and Its Association with CVDs), the authors should emphasize that wine is not completely safe and should describe the potential adverse effects of the excessive doses of the alcoholic content of wine including gastritis and potential ulceration. That is why the literature affirms the positive effects of long-term moderate doses of wine.   

11) In the abstract section, the authors are advised to avoid the general term “natural polyphenols” since the current review addressed only a few examples of them.  

12) More recent 2023 references are advised to be added to the current manuscript.

13) Under the type of article, please write “Review Article”.  

Minor editing of the English language is required.

Author Response

We would like to thank Reviewer #2 for the time and effort reviewing our manuscript and for providing valuable feedback. We have modified the manuscript according to the recommendations and incorporated all changes. Please find below a detailed, point-by-point response addressing your comments and concerns.

General Comments: The authors summarize evidence for the positive outcomes of resveratrol and other related polyphenols on the cardiovascular system. They list prior studies on the classification/biosynthesis/metabolism of polyphenols, stilbenes and CVDs, the effect of relevant flavonoids on CVDs, the association between CV mortality and polyphenol intake, and red wine consumption and CVDs. The authors provided one interesting figure that summarizes the primary sources and physiological effects of relevant polyphenols. Moreover, the authors provided one interesting table that summarizes the effect of various polyphenols on the cardiovascular system: selected randomized human clinical studies and meta-analyses. From this discussion, the authors suggest that several pre-clinical studies and clinical have demonstrated that the consumption of a polyphenol-rich diet, including moderate red wine consumption, may be beneficial in the prevention and treatment of CVDs.

Response: Thank you for your positive comment.

Comment 1: The provided sections read like narration for the evidence of discussed points without critical aspects/reflection points. At the end of each section, a take-home message is advised to be provided.

Response: Thank you for bringing up this important point. Now we have added a brief reflection to the end of the sections as appropriate. In certain instances, we felt that this is not possible based on the available literature or would only be a repetition of the already discussed facts.

Comment 2: In the introduction section, the authors are advised to elaborate on the novelty of the current review and how is it unique/updated relative to the previous reviews regarding the positive outcomes of resveratrol/related flavonoids on the cardiovascular system.

Response: Thank you for your remark. The novelty of this review article is to provide an extensive overview of favorable CV effects of polyphenols ranging from molecular mechanism to clinical results focusing on the most relevant natural polyphenols of red wine, including stilbenes and flavonoids. Moreover, we discuss in separate section CV outcome trials related to the effect of polyphenols on CV mortality, which is also a unique feature of the present work. We completed the section 1. (Introduction) according to your comment.

Comment 3: In order to attract the interest of more readers regarding the current review, the authors are advised to summarize the narration in section 3.1 (Mechanism of Action) with mechanistic schematic figure(s).

Response: Thank you for this constructive suggestion. We have now incorporated a new schematic figure (Figure 2) into section 3.1. of the manuscript.

Comment 4: In Table 1, the authors should add the route of administration, adverse events (if any), and retention in trial and other related flavonoids.

Response: Thank you for this comment. As per your suggestion, we have added the route of administration to Table 1. While we agree that providing further details in the table could assist readers to rapidly review individual studies, we also feel that adding further detailed information would significantly expand the Table negatively affecting readability and its primary aim to provide a brief and concise summary on the beneficial effects of flavonoids. Each of the source documents is cited at the end of our manuscript providing readers with the opportunity to review the details of each study at their convenience and according to their level of interest.

Comment 5: The authors should consider stratifying the narration on the effect on cardiovascular diseases into subsections of hypertension, atherosclerosis, myocardial infarction, and heart failure.

Response: Thank you for this suggestion. We have modified the manuscript according to your suggestion.

Comment 6: In the current review, the authors are advised to provide potential explanations of the reported controversy among several clinical trials with contrasting outcomes.

Response: Thank you for this comment. Although there would be a significant value in comparing the results of the various trials, we felt that given the differences in design, enrollment, substance dosing, it may be difficult and potentially misleading. In several instances it could significantly increase the length of the section without a profound clinical relevance. Therefore, we opted to prepare the paper as a narrative review rather than a systematic meta-analysis. We will plan to potentially complete such a paper in the near future.

Comment 7: The authors are advised to add a separate section on the reported adverse effects of resveratrol and other flavonoids, particularly, at higher doses. Moreover, their major interactions with cardiovascular medications should be described.

Response: This is an excellent recommendation that we completely agree with. With the update, we have added a section on the published adverse effects of polyphenols (Section 7: Adverse effects of relevant polyphenols). We believe this change made our manuscript significantly better and more complete.

Comment 8: The “conclusions” section should be replaced by “Conclusions and future directions”. The work should elaborate on future directions and limitations of previous studies and what is the next step to translate these findings to clinical settings.

Response: Thank you for this comment and recommendation; this is a very important point and may prove very useful for the readership of the journal. We have expanded “Conclusions” section according to your recommendation.

Comment 9: The authors should describe how they searched the literature, what are the keywords used, how many studies, and how did they select the proper studies (what are the inclusion/exclusion criteria for these studies)?

Response: Thank you for this comment. We have included the following section in the manuscript describing in detail the keywords and study inclusion criteria:
“The well-known search engine, PubMed was used to identify literature relevant to this review. The key words included the names of the included compounds, in addition to the terms “cardiovascular” or “cardiac” or “heart”. As the relevant full text articles were reviewed, the reference section of each manuscript was further reviewed to identify potentially relevant papers. Duplicate articles, as well as small studies with results not relevant to the current paper were excluded. Not aiming to provide a systematic meta-analysis, the format of narrative review was selected to summarize findings from relevant pre-clinical studies and human clinical trials”.

Comment 10: In section 6 (Red Wine Consumption and Its Association with CVDs), the authors should emphasize that wine is not completely safe and should describe the potential adverse effects of the excessive doses of the alcoholic content of wine including gastritis and potential ulceration. That is why the literature affirms the positive effects of long-term moderate doses of wine.   

Response: We completely agree with the Reviewer. We have modified Section 6 of the manuscript according to the suggestion emphasizing the importance of moderate alcohol consumption as well as describing some of the potential adverse effects of excessive use.

Comment 11: In the abstract section, the authors are advised to avoid the general term “natural polyphenols” since the current review addressed only a few examples of them.  

Response: Thank you for this suggestion. We have modified the abstract accordingly.

Comment 12: More recent 2023 references are advised to be added to the current manuscript.

Response: Thank you for this suggestion. We agree with the recommendation and further literature references have been included from 2023. These additional references are [48][116][119] [172].

Comment 13: Under the type of article, please write “Review Article”.  

Response: Thank you for this constructive comment. We have now added “a narrative review” to the title.

Thank you again for the Reviewer’s time and effort reading our paper and providing constructive feedback. We believe that the additions have improved the manuscript substantially.

Sincerely yours,

László Czopf MD, PhD

Reviewer 3 Report

Manuscript No biomedicines-2621835

Resveratrol and Beyond: The Effect of Natural Polyphenols on the Cardiovascular System” for biomedicines

Comments:

1.      Minor. Please standardize the font size in the text.

2.      Paragraph 3.1. Please clearly link the changes in the level of pro-inflammatory cytokines, COX-1 and COX-2, with the activity of the vascular endothelium in your description. At the moment, these are just loose statements in this paragraph.

3.      Please also refer to the role of polyphenols in regulating the tension of vascular walls. What effect do they have on individual layers of vessel walls and, consequently, on CVDs?

Author Response

We would like to thank Reviewer #3 for their time and effort thoroughly reading our manuscript. Please find below the point-by-point responses to the reviewer’s comments and concerns:

Comment 1: Minor. Please standardize the font size in the text.

Response: Thank you for your comment. The font size has been standardized throughout the manuscript.

Comment 2: Paragraph 3.1. Please clearly link the changes in the level of pro-inflammatory cytokines, COX-1 and COX-2, with the activity of the vascular endothelium in your description. At the moment, these are just loose statements in this paragraph.

Response: Thank you for this constructive comment. We have modified Section 3.1 in the manuscript according to your recommendation (Page 9).

Comment 3: Please also refer to the role of polyphenols in regulating the tension of vascular walls. What effect do they have on individual layers of vessel walls and, consequently, on CVDs?

Response: Thank you for this comment. We have edited Section 3.1 according to your suggestion (Page 11).

We would like to thank the Reviewer again for their time and effort, the modifications based on these comments indeed improved the manuscript significantly.

Yours sincerely,

László Czopf MD, PhD

Round 2

Reviewer 2 Report

The authors have adequately addressed the raised comments. Thanks!

Minor editing of the English language is required.